# Rare Variants Residing in Novel Cis-Acting Element in Visual System Homeobox 1 and Their Contribution in the Pathogenesis of Keratoconus

Farhan Khashim Alswailmi [1], Rida Khursheed Malik [2], Mujeeb Ur Rehman Parrey [3], Abdul Rauf Siddiqi [2], Shaik Karimulla [1], Abdulkareem A. Alanezi [4], Raheel Qamar [5,6], Maleeha Azam [2,*] and Ashfaq Ahmad [1,*]

[1] Department of Pharmacy Practice, College of Pharmacy, University of Hafr Al Batin, Hafr Al-Batin 39911, Saudi Arabia

[2] Translational Genomics Laboratory, Department of Biosciences, COMSATS University Islamabad, Islamabad 45600, Pakistan

[3] Faculty of Medicine, Northern Border University, Arar 91431, Saudi Arabia

[4] Department of Pharmaceutics, College of Pharmacy, University of Hafr Al Batin, Hafr Al-Batin 39911, Saudi Arabia

[5] Pakistan Academy of Sciences, Islamabad 45600, Pakistan

[6] Science and Technology Sector, ICESCO, Rabat 10104, Morocco

* Correspondence: malihazam@gmail.com (M.A.); ashfaqa@uhb.edu.sa (A.A.)

**Abstract:** (1) Background: The visual system homeobox 1 (VSX1) may contribute to the incidence of keratoconus (KC) in different populations. The present study investigated the role of VSX1 in autosomal recessive Pakistani families and sporadic KC patients using in silico analysis of the rare variants for the identification of the cis-acting elements in *VSX1*; (2) Methods: Mutation analysis of *VSX1* was undertaken using Sanger sequencing of samples from seven KC families and 100 sporadic patients. In silico analysis of the rare variants and identification of cis-acting elements was determined using Human Splicing Finder (HSF), ESE finder, RESCUE-ESE and through Exon- Identity Element (EIEs) prediction software suits, combined with various algorithms to identify the effect of variations in splicing motifs; (3) Results: Screening of *VSX1* did not reveal any novel mutation in the KC panel, but a synonymous polymorphism rs12480307 (c.546A>G; p.Ala182Ala) in exon three and 3′UTR rs76499395 (c.*496A>G) were observed in two separate probands. These polymorphisms were not found in any of the sporadic KC cases or 100 ethnically matched control samples. The analysis of these rare variants revealed a plausible role for these two single nucleotide polymorphisms (SNPs) in KC development through the identification of novel cis-acting elements, an exonic splicing enhancer element (ESE) and binding motifs for two micro RNAs, miRNA-203 binding and hsa-miR-3938, in the VSX1 gene structure; (4) Conclusions: Rare genetic variations in the *VSX1* were found to have a potential contribution to KC development.

**Keywords:** keratoconus; VSX1; rare variants; cis-acting elements; miRNA

## 1. Introduction

Keratoconus is cone shaped and occurs as bilateral corneal dystrophy, which is expressed as a progressive thinning of the corneal layers of stroma [1]. The condition has a multifactorial origin and is progressive but non-inflammatory. The initial corneal thinning leads to irregular astigmatism, a prominent level of myopia and a severe refractive error in the patient [2,3], resulting in decreased visual acuity. At the initial stages of the disease, there are no obvious symptoms other than a mild loss of visual acuity. As the disease progresses, severe signs start to appear, such as Fleischer's ring, corneal protrusion, Vogt's striae and Munson [4]. Both eyes are affected in an asymmetric manner [5]. Globally the prevalence of keratoconus is 1 out of 2000 individuals, which varies with ethnicity having a prevalence rate in Caucasian populations of between 8.8–54.5 per 100,000 individuals [1], in

a Malaysian population of 1 per 100 persons [6] and in an Iranian population 760 patients out of 100,000 individuals which is much higher than the prevalence of KC in the Western world population [7]. The same disease was studied in the United States for 48 years, and the data collected over this time reported the disease prevalence to be 54.5 patients per 100,000 individuals [8]. A study reported in 2017 showed the prevalence of disease in children in Saudi Arabia to be 4.8% [9].

Manifestation of KC usually starts in the second decade of life and progresses in severity over the third and fourth [10], while the prevalence of this disease is higher in males compared to females in certain populations [11,12]. The pattern of the disease prevalence is more sporadically oriented than familial linked, but a positive family history does contribute to the risk of the disease in the offspring. In familial linked cases, a history of the presence of inherited autosomal dominant mode is frequent when compared to X-linked or recessive inheritance [13,14]. The exact etiological cause of KC is unclear due to the varying prevalence of the disease in different races and families, and it occurs sporadically. Nonetheless, bearing in mind KC is a multifactorial disease, it can be speculated that both the environmental as well as genetic factors play their individual and specific roles in the pathogenesis of the disease [15].

Linkage analysis of the Keratoconus patient population of different ethnicities and Genome-wide association studies (GWAS) have indicated the involvement of various single nucleotide polymorphisms (SNPs) and loci in the pathogenesis of the disease [16], which includes matrix metalloproteinase (*MMP-9*), inhibitors of metalloproteinase 3 (*TIMP-3*), lysyl oxidase (*LOX*), collagen type VI alpha-1 chain (*COL6A1*), hepatocyte growth factor (*HGF*), superoxide dismutase (*SOD*), crumbs homolog 1 (*CRB1*), zinc finger e-box binding homeobox 1 (*ZEB1*) and visual system homeobox 1 (*VSX1*). Among all these genes, proteins and factors, VSX1 is involved in human ocular development [17] but it is also expressed in embryonic craniofacial tissue [18], the inner layer of the retina, as well as corneal tissue in the adult [19].

Despite the number of associated genes, the role of genetic deficits in the pathogenesis of keratoconus has not been explored in Pakistani families or the sporadic population. Moreover, bioinformatic information regarding the role of genetic variants and their impact on gene/protein function is severely lacking. The present study is designed to explore the contribution of the *VSX1* genes in the recessive form of the disease in Pakistani KC patients and to perform bioinformatic analysis on identified variants to explore their potential contributions to KC development.

## 2. Materials and Methods

### 2.1. Sample Collection and DNA Isolation

Seven (07) autosomal recessive KC families (Figure 1) and one hundred (100) sporadic KC cases were recruited locally from hospitals in Islamabad, Pakistan. Blood sampling of the Keratoconus patients was performed irrespective of age, sex or occupation (males = 64, females = 36). The average age of onset of KC in the current cohort was 20 years. Patients with a positive family history of KC and those who experienced eye rubbing due to allergies, together with a family history of KC, were recruited into the study. Exclusion criteria from the study consisted of eye trauma and other corneal pathologies in patients where keratoconus was secondary to another primary disease, infection or those who had excessive eye rubbing.

All the recruited KC patients underwent corneal examination. This comprised an examination to determine the average Orb scan measurements, which evaluates the curvature of both anterior and posterior corneal layer. In the KC patients, it was determined to be 438.6 ± 47.7 μm (normal range~550 μm), which suggested a thinning of the cornea. Based on the measurement of corneal thinning, corneal pachymetry showed that the patients had an average thickness of 462.5 μm ± 8.0 (normal range~554.9 μm ± 7.4). The curvature of the anterior surface was measured using Keratometry, and the patients had a curvature of 52.5 ± 8.8 D (normal range 7.2 ± 0.8 D). Age, sex and occupation matched 100 healthy

control individuals (males = 62, females = 38) who were also randomly recruited into the study as control subjects for the KC cohort, and these controls were naïve for any eye disease or inherited disease and had no major health issues. Once all the legal requirements and written consent had been obtained, blood samples were taken from all KC patients and control individuals. The present study complied with the Helsinki declaration, while ethical and scientific approval was obtained from the Ethics Committee/Institutional Review Board Department of Biosciences, COMSATS University Islamabad, Pakistan.

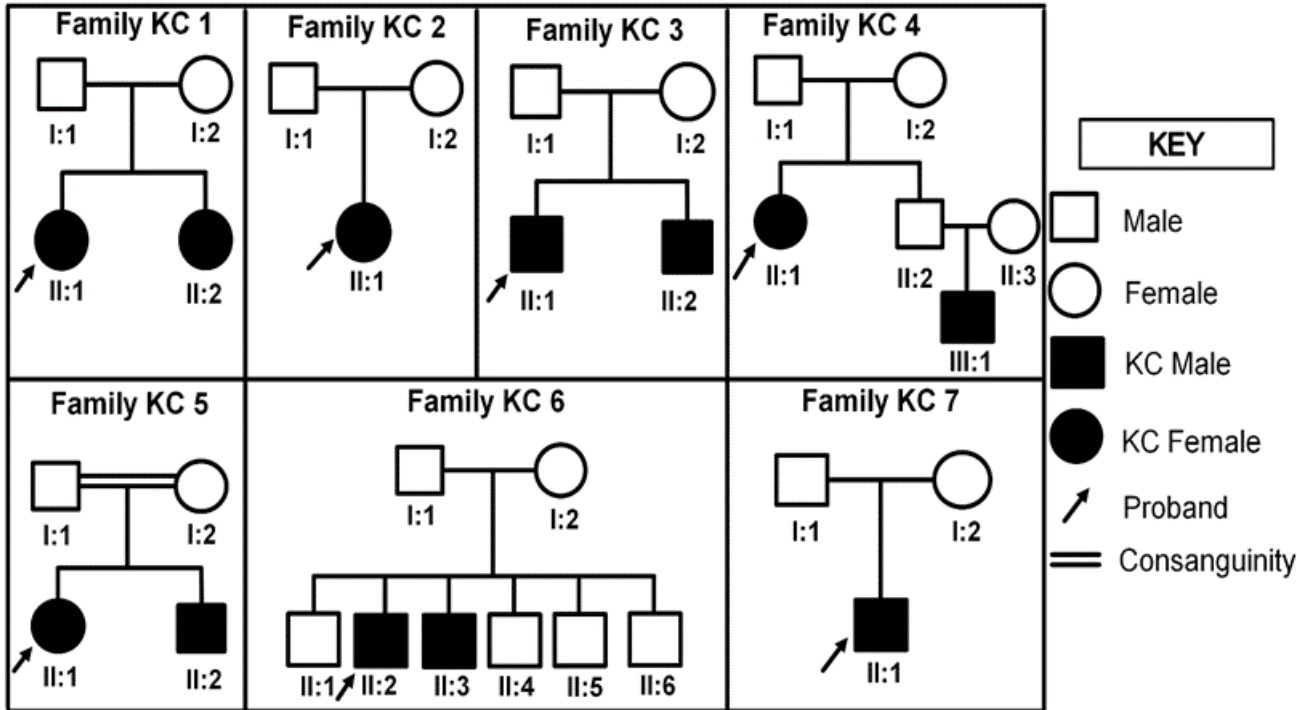

**Figure 1.** Pedigrees for seven autosomal recessive KC families. In the pedigrees white circles represent unaffected females whereas white squares represent unaffected males. Filled symbols represent affected individuals. Double lines indicate consanguineous marriage between parents.

Genomic DNA was extracted from the plasma lymphocytes of the patient using the salting-out method as described previously [20]. Briefly, DNA extraction consisted of lysis of red blood cells (RBC) with erythrocyte lysis buffer (ELB) followed by the lysis of white blood cells (WBC) with Tris-Natrium chloride-ethylenediamine tetra acetate buffer (TNE), SDS and Pronase E. These samples were incubated overnight at room temperature, and DNA was salted out using NaCl salt and ethanol solution, which later then precipitated and re-suspended in TE buffer. This DNA was stored at −20 °C for further study.

*2.2. VSX1 Sequencing*

The exons and exon-intron boundaries of *VSX1* in 9 KC family probands (Figure 1) and 100 sporadic cases were sequenced. Amplification of the probands from the KC sporadic cases was performed using the primers and conditions given in Table 1. The reaction consisted of mixing 50–70 ng of genomic DNA, 0.5 mM dNTPs, 1X ammonium sulfate Taq buffer, 2.5 U Taq DNA polymerase and GIBCO® water. Thermocycling was performed at an initial denaturation at 95 °C for 5 min, immediately followed by 35 cycles of denaturation at the same temperature (95 °C) for 30 s; primer annealing was conducted for 30 s at the respective temperatures given in Table 1 and chain extension at 72 °C for 30 s. The final extension was conducted at temperature 72 °C for 7 min.

**Table 1.** Sequencing primers for *VSX1* exons with amplification conditions.

| Exons | Primer Sequence | Annealing (°C) | Product Size (bp) | MgCl$_2$ (mM) | Primer (µM) |
|---|---|---|---|---|---|
| Exon 1_1 | F: 5′-TTTCGAGGGACAGGCAGAC-3′<br>R: 5′-AGGTCCGTGATGGCGAAG-3′ | 60 | 449 | 2 | 0.4 |
| Exon 1_2 | F: 5′-TGCTTGCTAAGGAACCATGAC-3′<br>R: 5′-TCAGAGCCTAGGGGACAGG-3′ | 61 | 489 | 2.5 | 0.32 |
| Exon 2 | F: 5′-AATGCTGGCTCATACTGTAAAC-3′<br>R: 5′-AACCAGGAAACCACTGGG-3′ | 58 | 327 | 2.5 | 0.4 |
| Exon 3 | F: 5′-AGCAGAGGAAGCAGGCAC-3′<br>R: 5′-CTATGCAAAGGGAGCGTG-3′ | 58 | 332 | 2.5 | 0.4 |
| Exon 4 | F: 5′-ATCATGCTCGGGAGAGAAG-3′<br>R: 5′-TTGCTTTGCTTTGGAAATG-3′ | 58 | 391 | 3 | 0.4 |
| Exon 5_1 | F: 5′-CCCCAGAGATAGGCACTGAC-3′<br>R: 5′-TGCCAGTGAGGAATATGCAC-3′ | 58 | 470 | 3 | 0.4 |
| Exon 5_2 | F: 5′-GCAGGAGACCAAGAAAGTGC-3′<br>R: 5′-CTCAAATGATGCCCAGCAG-3′ | 58 | 416 | 2.5 | 0.4 |
| Exon 5_3 | F: 5′-ATGCCACTTGCTTTAAGAGG-3′<br>R: 5′-TGCAGAAACGACTAGAGTATGG-3′ | 58 | 464 | 3 | 0.4 |
| Exon 5_4 | F: 5′-TACCTTGAACTTGGCCTTGG-3′<br>R: 5′-TGGCTGGGATCAGAGATAGTG-3′ | 58 | 391 | 2.5 | 0.4 |

*VSX1*, Visual system homeobox gene; F, forward primer; R, reverse primer.

### 2.3. In Silico Analysis

The exon three sequence of *VSX1* was analyzed for the presence of potential cis-acting splicing-regulatory elements (SREs) using Human Splicing Finder (HSF), Exonic splice enhancer (ESE) element finder and through Exon-Identity Element (EIEs) prediction software. All three software have been widely used and cited to predict SREs. HSF predicts the effect of a mutation in an exonic region on splicing by combining twelve different algorithms, which include position weight matrices, maximum entropy and motif comparison algorithms [21]. The ESE finder identifies and predicts the ESE elements by scanning the subject's exonic sequence for the presence of binding motifs of four SR proteins; the binding motifs are searched on the basis of position-specific scoring matrices based on frequency weighting values derived from the alignment of winner sequences [22]. Zhang et al. [23] proposed a model for the identification of cis-acting SREs which scans the subject's sequence for exon-identity elements (EIEs) and intron-identity elements (IIEs). The program identifies EIEs and IIEs based on asymmetry in the nucleotide frequency and distribution compared to the rest of genomic DNA. The asymmetry exhibited by the SREs would have developed due to evolutionary selection pressure over the SREs, but the remainder of the nucleic acid shows a symmetric frequency of distribution of bases. Another program used was RESCUE-ESE, which uses a position weight matrix-based strategy validated experimentally and known as ESE sequences [24].

The 3′UTR sequence of *VSX1* was checked using PITA (Probability of Interaction by Target Accessibility) software. PITA is a widely used algorithm for screening miRNA binding regions in the 3′UTR [25]. PITA has devised a model which successfully predicts miRNA binding sequence motifs based on the difference between free energies of unbound duplexed target mRNA with a hidden miRNA site and those of an open mRNA with a binding site accessible to the subject miRNA. The difference between two free energies is termed ΔΔG; the lower the ΔΔG value, the better will be the binding of miRNA with the target.

The 3′UTR of *VSX1* was also analyzed through the miRanda algorithm. The miRanda algorithm functions, such as the Smith-Waterman alignment algorithm, to search for potential miRNA binding sequences, but instead of aligning the matching nucleotides (A-A and U-U), it searches for the target sequence based on the complementarity of base pairs (A=U or G≡C). miRanda is a widely cited algorithm, it searches for complementarity matches between miRNAs and 3′UTR targets through alignment generated by dynamic programming [26,27].

## 3. Results

### 3.1. Pedigree Analysis

The detailed interview of family members and review of previous medical records of keratoconus patients revealed the families as having a positive family history of the disease in different branches of the families. Sample collection from available affected members of the families along with their parents, followed by a pedigree analysis of the phenotype, showed the mode of inheritance to be autosomal recessive in all the seven families sampled.

### 3.2. Screening of VSX1

Sequencing of the *VSX1* gene did not show any mutations in the seven keratoconus families. However, a single nucleotide polymorphism (SNP), rs12480307 (c.546A>G; p.Ala182Ala) in exon three with minor allele frequency (MAF) of G = 0.252, was present homozygously in the proband of a family KC2 (Figures 1 and 2A) and another 3′UTR rs76499395 (c.*496A>G) with MAF = 0.02 SNP, was present heterozygously in the proband of family KC7 (Figures 1 and 2B). Sequencing of *VSX1* did not show any mutations in the one hundred sporadic keratoconus cases.

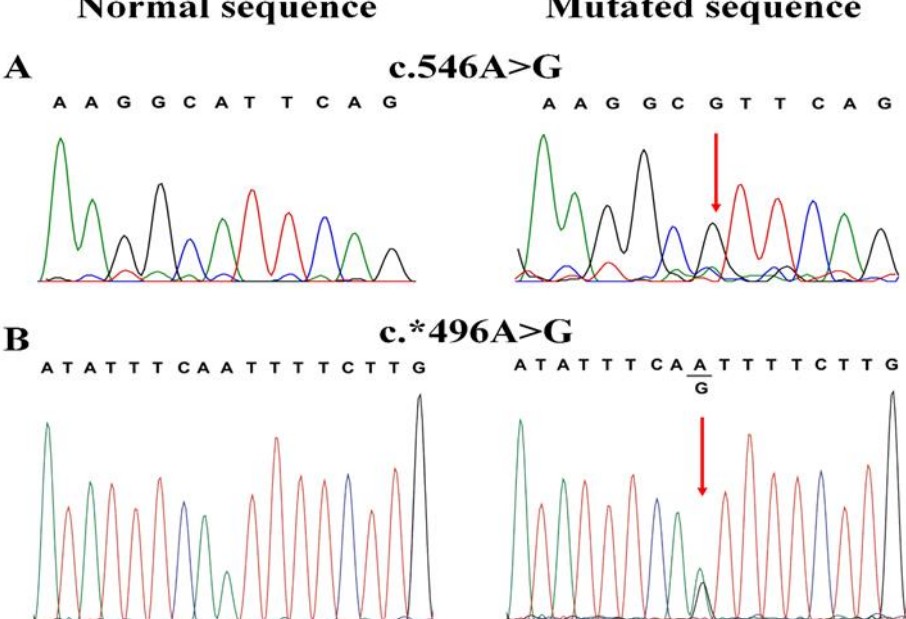

**Figure 2.** Sequence chromatogram of observed variants. Upper panel (**A**) represents normal and mutated sequence for rs12480307 (c.546A>G) while lower panel (**B**) is of rs76499395 (c.*496A>G) polymorphism.

### 3.3. In Silico Analysis of c.546A>G

The rs12480307 (c.546A>G; p. Ala182Ala) sequence lies in the third exon of the *VSX1*, which is close to a splice site; therefore, the SNP, along with flanking sequences, was analyzed to determine whether these contained any ESE or ESS sequences. It was observed that the *VSX1* SNP c.546A>G was altered and broke an ESE element lying in the third exon of the *VSX1*. This has been observed using three different splicing analytical tools used in this study. HSF predicted an ESE element starting from position *VSX1* c.546, spanning over

the next seven 3′ nucleotides with the sequence ATTCAGC. HSF predicted that in terms of SNP c.546A>G, the ESE element would be altered and broken. RESCUE-ESE also predicted the same ESE at position c546-552 of *VSX1*. The EIE identifier also predicted an exonic SRE at the same site (Table 2). The ESE finder identified a strong SRSF5 binding motif ATTCAGC (*VSX1* c546-552) with a high binding score of 3.07, well above the threshold score of 2.67. In the case of SNP c.546A>G SRSF5, binding was lost for SRSF 5, but the mutant motif GTTCAGC showed a binding affinity with the SRSF2 protein (Table 3). Figure 3 illustrates the pre-mRNA analogs of the ESE elements, *VSX1* Wild Type c.546A-c.553G (ATTCAGCG) and *VSX1* SNP c.546A>G-c.553G (GTTCAGCG), bound to SRSF5 as revealed by docking. The wild type of ESE segment, c.546A-c.553G (ATTCAGCG), binds right in the binding pocket of the SRSF 5 protein, whereas the SNP segment, c.546A>G-c.553G (GTTCAGCG), did not bind well except for its initial 5′ region, docking nearly 3–4 Å away from the binding pocket of the SRSF 5 molecule. This would affect the splicing of the corresponding exonic region.

**Table 2.** Splicing analysis of the exon 3 sequence of the *VSX1* and its SNP c.546A>G, effect on the ESE site the formation of splicing regulatory complex.

| Predicted Signal | Prediction Algorithm | cDNA Position | Interpretation |
|---|---|---|---|
| ESE Site Broken | 1. ESE-Finder-SRp40<br>2. RESCUE ESE Hexamers<br>3. EIEs from Zhang et al. [23] | G G C A T T C A G C G A G<br>cDNA 543 545 547 549 551 553 555 | Alteration of an exonic ESE site. Potential alteration of spllicing |

**Table 3.** Variation in ESE-SRSF protein binding properties between *VSX1* Wild Type c.546A as compared to *VSX1* SNP c.546A>G analyzed using the ESE finder.

| Type | SRSF Binding Properties | | | |
|---|---|---|---|---|
| | Biding Motif | SRSF Protein | Binding Score | BS Threshold |
| *VSX1* Wild Type | c.546A-c.553G ATTCAGCG | SRSF5 | 3.07 | 2.67 |
| *VSX1* SNP | c.546A>G-c.553G GTTCAGCG | SRSF2 | 3.20 | 2.38 |

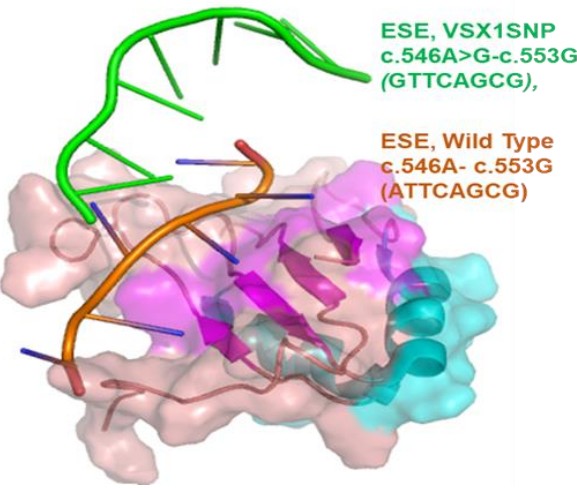

**Figure 3.** ESE elements, *VSX1* Wild Type c.546A-c.553G (ATTCAGCG) shown in brown and *VSX1* SNP c.546A>G-c.553G (GTTCAGCG) showing green docked to the SRSF 5 binding pocket. Notice the Wild Type ESE element binds well into the SRSF 5 binding pocket, while the green one binds away, excepting its 5′ region.

Both PITA and miRANDA predicted a miRNA binding site at *VSX1* 3′UTR c.*496 position (Table 4). The wildtype (WT) sequence was found to show strong binding with miRNA hsa-miR-203, and the binding site spanned across c*475 to c.*496 in *VSX1* 3′UTR wildtype (*VSX1* 3′UTR (WT) (Figure 4A). PITA estimated ΔΔG values for miRNA-203 binding with *VSX1* 3′UTR (WT) as −9.34, the seed region spanned over eight base pairs with all bases pairing up with target mRNA as one wobble pair. The miRNA-203 showed a minimum free energy of −15–50 kcal/mol with a binding score of 150, as determined by miRANDA. Conversely, the *VSX1* 3′UTR mutant type (MT), c.*496 A>G lost the binding site for miRNA-203, but instead, c.*496G was revealed as the binding site for hsa-miR-3938 from c.*496-c.*514 with a binding affinity of −12 kcal/mol and a binding score of 142 (Table 4; Figure 4A,B).

**Table 4.** Variations in binding properties of miRNA-203 and miRNA-3938 with *VSX1* 3′UTR (WT) as predicted by miRANDA and PITA.

| miRNA | Wild and Mutant Type Sequences | miRANDA | | | PITA | |
|---|---|---|---|---|---|---|
| | | Binds | Binding Free Energy (kcal/mol) | Score | Binds | ΔΔG |
| miRNA-203 | VSX1 3′UTR(WT) c* 496 A | ✓ | −15.50 | 150 | ✓ | −9.35 |
| | VSX1 3′UTR(MT) c* 496 A>G | X | 0 | 0 | X | 0 |
| miRNA-3938 | VSX1 3′UTR(WT) c* 496 A | X | 0 | 0 | X | 0 |
| | VSX1 3′UTR(MT) c* 496 A>G | ✓ | −12.00 | 140 | ✓ | −7.65 |

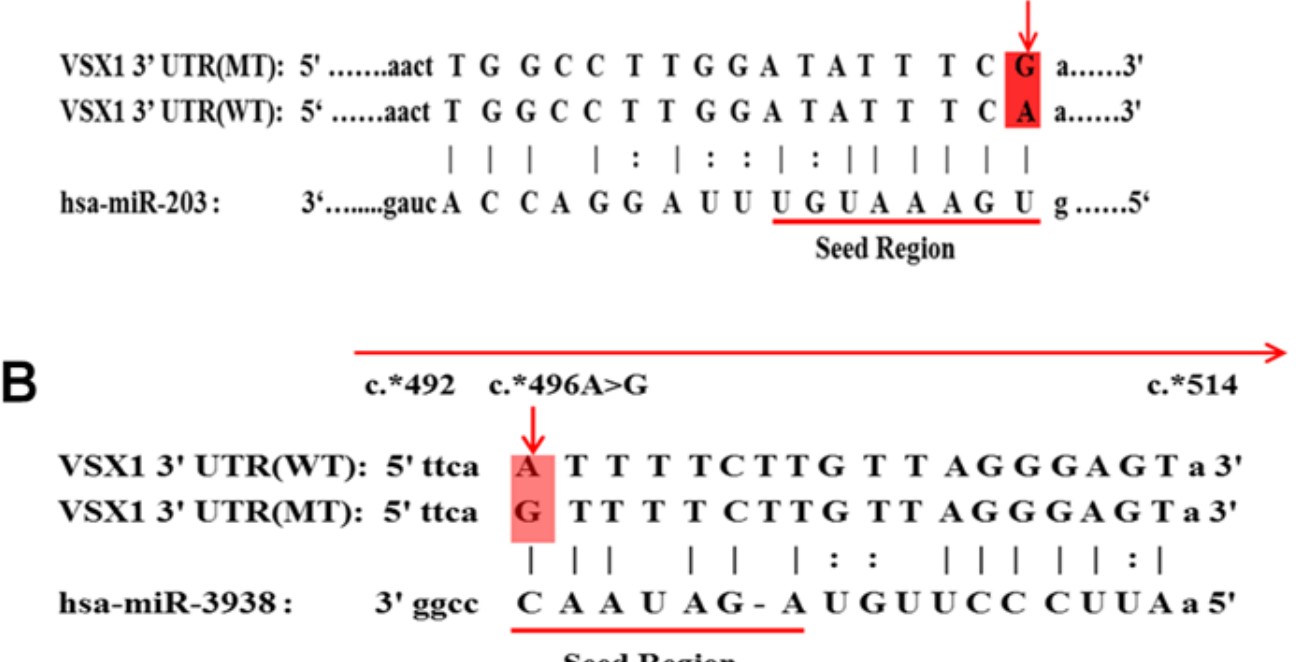

**Figure 4.** hsa-miRNA-203 binding site on *VSX1* 3′UTR(WT). (**A**) c*475-c*496 as predicted by PITA and miRANDA. (**B**) In the case of *VSX1* 3′UTR(MT) c*496 A>G, hsa-miRNA-203 there is no binding.

## 4. Discussion

This study was designed to investigate the contribution of *VSX1* in autosomal recessive genes in Pakistani keratoconus families compared with that in subjects appearing as sporadic cases. This study was also set to perform an in silico analysis of the rare variants to allow for the identification of cis-acting elements in *VSX1*. In the present study, screening of the previously reported KC-associated gene *VSX1* revealed a potential role for rare variants which might contribute to KC manifestation. This was achieved through the identification of novel cis-acting elements in *VSX1* based on bioinformatic analysis of the gene.

The results of the *VSX1* screening in the current panel were not in agreement with those reported earlier, where most of the cases carried mutations in the coding part of the gene and were reported for the dominant form of the disease. However, in the present cohort of autosomal recessive KC families, screening of *VSX1* found no novel or recurrent mutation except for two SNPs rs12480307, a synonymous variation, and a 3′UTR rs76499395. A reasonable number of studies using *VSX1* mutational screening reported *VSX1* exon two as a mutation hot spot in KC cases, but the SNPs observed in two patients in the present study lie in the third exon and 3′UTR sequences. The results of the sporadic cases were also like that of the familial probands, exhibiting no *VSX1* mutations. Regarding the mutation identification, the findings of the present study were consistent with those undertaken in Caucasian [28], Czech [29], Italian [30], Slovenian [31], South Indian [32] and North Indian [33] populations, in which no coding sequence *VSX1* mutations were found in KC cases while ignoring very rare variants and their possible role in disease manifestation. Moreover, the expression analysis of *VSX1* in human and mouse corneas also indicated contradictory findings. In humans [19], *VSX1* expression in the adult cornea was not detected, but it was found to be higher in retinal tissues [34].

Although polymorphisms in different genes have been reported to be one of the risk factors in the pathogenesis of KC worldwide, a number of studies have reported that the association of SNPs with KC is low but did not investigate the underlying gene regulatory mechanisms, SREs and miRNAs which might be implicated in such associations. The identification of the ESE element (containing c.546A>G SNP) and a miRNA binding site at *VSX1* 3′UTR c.*496 position has highlighted the functional aspects of SNPs on cis-acting elements and their consequent effect on gene expression. Among the regulatory sequences, splicing is determined by a regulatory complex comprising a number of snRNP (small nuclear ribonucleo proteins) often named after their small nuclear RNA (snRNA) segments U1, U2, U4, U5 and U6 [35]. The role of snRNPs is to identify and bind at specific splice donor, branch and acceptor sites characterized by GU, polypyrimidine and AG nucleotides lying in the intronic region. The recruitment of these proteins is initiated by specific serine/arginine-rich (SR) proteins, the SRSF or SR (specific serine/arginine-rich) proteins. These start the recruitment process of snRNPs only after binding at specific exonic splicing enhancer (ESE) motifs lying in the flanking exonic regions of the subject introns. Twelve different SRSF proteins are known to bind at their specific ESE elements in pre-mRNA to initiate the splicing regulatory complex. Once the complex is formed, the spliceosome splices out the intronic region joining the two neighboring exons to form the mature mRNA for translation [36,37]. The current findings, therefore, clearly indicated that the *VSX1* c546-552 was occupied by a highly putative ESE site, and in the case of the SNP c.546A>G, the homozygously present G allele, the ESE loses its affinity for a specific SR protein. This alteration could have consequences, such as exon skipping, for example, leading to the exclusion of an entire exon from *VSX1* pre-mRNA transcript and consequently affecting gene expression. However, these findings need further investigation and validation through in vitro functional analysis.

MiR-203 is a widely studied miRNA for mediation in repressing several types of disorders, including multiple types of malignancies and cancer [38,39]. miRNA-203 has been found to suppress cell proliferation and the metastasis of several types of colorectal cancers by targeting Eukaryotic initiation factor 5A2 (EIF5A2) [40]. On the other hand, miRNA-3938 not only plays a role in cancers but has also been reported to affect macrophage

cell development [41]. The loss of binding of miRNA-203 with *VSX1* 3′UTR c*496A>G (MT) and the binding of miRNA-3938 could be implicated in KC; however, the presence of variant c*496A>G heterozygously in KC patients could indicate a penetrance/dosage effect because one of the parents that carried the variation did not exhibit the phenotype, whereas disease manifestation was seen in offspring. The dosage effect of *VSX1* variants in KC families has been observed in one previous study [42], which reported the Q175H *VSX1* variant as being pathogenic with incomplete penetrance in the patient but not in the carrier mother. The genetic variants demonstrating such reduced penetrance have been reported to involve some modifier genes or environmental factors for disease presentation. It is also possible that the variant c*496A>G requires some genetic modifier or environmental factor to be pathogenic.

Like the previous studies by Aldave et al. [28], Liskova et al. [29], Tang et al. [30], Stabuc-Silih et al. [31], Verma et al. [32] and Tanwar et al. [33] on *VSX1* screening in different ethnicities worldwide (Table 5), there seems a similarity in previous studies regarding the cohort selection and sample size. However, none of the studies were able to solve a complete cohort involving sporadic as well as familial cases. Most of the studies reported polymorphic variations, including synonymous as well as non-synonymous and intronic variations; however, due to the absence of in silico analysis, the exact role of the non-coding variations and microRNA was not investigated in previous studies. These contradictory results, therefore, highlight and suggest that KC cohorts, including sporadic and familial cases belonging to different ethnicities, to be genetically screened using state-of-the-art techniques, such as genome and exome sequencing to identify the exact genetic cause. These limitations in the current and previous studies, therefore, suggest thorough investigations involving larger KC cohorts and genome sequencing of KC patients. Another limitation of the current and previous findings is the lack of in vitro and in vivo functional characterization of identified genetic variants in KC patients. Regarding the cis-acting element and their predicted association with microRNA in the current study, though functional analysis could not be performed, however number of studies have reported the role of microRNA-203 in the increased inflammatory response [43]. The role of inflammation in KC pathogenesis is well defined [44,45]; therefore, the functional validation of the role of microRNAs in KC manifestation is suggested in future KC investigations.

**Table 5.** Comparison of the studies conducted on same pattern on different ethnicity, study type and cohort size, techniques, identified variants and number of patients.

| Ethnicity | Study Type and Cohort Size | Isolated/Familial Cohort | Technique | Identified Variant(s) | No.of Patient(s) | Variant Type | Reference |
|---|---|---|---|---|---|---|---|
| Caucasian | Association study. 100 unrelated KC Patients | Isolated cohort | Sanger Sequencing | p.Asp144Glu<br>p.Ser6Ser<br>p.Pro58Pro<br>p.Arg131Ser<br>p.Ala182Ala | 1<br>4<br>2<br>1<br>51 | Non-synonymous<br>Synonymous<br>Synonymous<br>Non-synonymous<br>Synonymous | [28] |
| Caucasian+Asian +African | 85 Probands | Familial | Sanger Sequencing | p.D144E<br>c.504-10G>A<br>504-24C>T | 1<br>1<br>3 | Non-synonymous<br>Intronic<br>Intronic | [29] |
| Whites+ Hispan-ics+Others | Association study with 77 unrelated KC Patients and 75 families | Isolated co-hort+familial cohort | ARMS-PCR and RFLP | p.H244R<br>p.L159M | 3<br>5 | Non-synonymous<br>Non-synonymous | [30] |
| Slovenian | 113 patients with sporadic and familial KC | Isolated co-hort+familial cohort | Sanger Sequencing | p.S6S<br>p.A128A<br>p.D144E<br>504-24C.T<br>627+23G.A | 21<br>35<br>1<br>0<br>44 | Synonymous<br>Synonymous<br>Non-synonymous<br>Intronic<br>Intronic | [31] |
| Indian | 117 sporadic cases of keratoconus | Isolated cohort | Sanger Sequencing | p.A182A<br>c.627+23G>A<br>c.627+84T>A<br>c.504-24C>T | 7<br>3<br>9<br>7 | Non-synonymous<br>Intronic<br>Intronic<br>Intronic | [32] |
| Indian | 50 sporadic cases | Isolated cohort | Sanger Sequencing | p.R217H<br>p.P237P | 1<br>3 | Non-synonymous<br>Synonymous | [33] |

## 5. Conclusions

The present study is one of the first to report the pathophysiological involvement of genetics role of cis-acting elements and microRNA in KC, where the role of rare variants of *VSX1* was highlighted through the identification of novel cis-acting elements and their contribution to KC development as guided by in silico analysis. The identification of miRNA binding sites in the case of wild type and variant sequences has given new insight into disease causation. In vitro experimentation is required further to validate the current findings.

**Author Contributions:** Concept, F.K.A., M.A. and A.A.A.; methodology, R.K.M.; validation, F.K.A. and M.U.R.P.; investigation, F.K.A. and R.Q.; formal analysis, M.U.R.P., A.R.S. and R.K.M.; software, A.R.S., M.A. and A.A.; data curation, S.K. and F.K.A. resources, M.A., F.K.A. and A.A.A.; writing—original draft preparation, M.A., R.K.M. and F.K.A.; writing—review and editing, A.R.S., A.A.A. and S.K. project administration, F.K.A. and A.A.A.; supervision, A.A.A. and M.A.; funding acquisition, F.K.A. All authors have read and agreed to the published version of the manuscript.

**Funding:** The current study acknowledges the Deanship of Scientific Research, University of Hafr Al batin for research grant project No. 0015-1443-S.

**Institutional Review Board Statement:** The project was approved by the Ethics Review Board, Department of Biosciences (CUI-Reg/Notif-452/20/526), COMSATS University Islamabad. The study was conducted in accordance with the Declaration of Helsinki.

**Informed Consent Statement:** Informed consent was obtained from all subjects involved in the study.

**Data Availability Statement:** All data generated in this study is included as a supplementary file.

**Acknowledgments:** The authors thank all the participants for their contributions to the study.

**Conflicts of Interest:** The authors declare no conflict of interest.

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
