# Peer review of "Rare Variants Residing in Novel Cis-Acting Element in Visual System Homeobox 1 and Their Contribution in the Pathogenesis of Keratoconus"

_applsci, doi:10.3390/app13084888_

Round 1

Reviewer 1 Report

The causative relationship is probed between Visual system homeobox 1 (VSX1) and keratoconus (KC) in autosomal recessive Pakistani families and sporadic KC cases. For the purpose in silico analysis is performed of the rare variants for the identification of cis-acting elements in VSX1.

The sample cohort consisted of seven KC families and 100 sporadic cases for which mutation analysis of VSX1 is performed via Sanger sequencing. The criteria and protocol of selection of individuals for the study is thoroughly explained and motivated. The in silico analysis of the rare variants and identification of cis-acting elements was done with wide range of relevant tools: Human Splicing Finder (HSF), ESE finder, RESCUE-ESE and through Exon-Identity Element (EIEs) prediction software suits, combined with various algorithms for identification of the effect of variations on splicing motifs.

The results of VSX1 screening disagreed with those of earlier studies where most of the cases carried mutations in coding part of the gene and were therefore thought to be the dominant form of the disease. Instead, in the present cohort, the autosomal recessive KC families showed no novel or recurrent mutation except for two SNPs rs12480307, a synonymous variation, and a 3'UTR rs76499395. These polymorphisms were not found in any of the sporadic KC cases and 100 ethnically matched control samples. The in silico analysis identified novel cis-acting elements, an exonic splicing enhancer element (ESE) and binding motifs for two micro RNAs miRNA-203 binding and hsa-miR-3938, in the VSX1 gene structure. Such findings indeed suggest a possible role in KC development.

 Thus the present study provides an original contribution on the pathophysiological role of genetics in KC where the role of rare variants of VSX1 is emphasized through identification of novel cis-acting element and their role in KC development guided by in-silico analysis. The identification of miRNA binding sites in case of wild type and variant sequence has given new insight into disease causation.

Overall it is relevant study, but some changes and clarifications will enhance it further:

- It will be good to compare (in terms of advantages and limitations) the cohort size and structure (number of patients and how they are distributed among the studied categories) in this study to the cohort size and structure in previously quoted studies on the topic (e.g. references 28-33). Such critical analysis may provide partial explanation of the discrepancies between the results reported by different teams.

- The English spelling and grammar need thorough improvement which will increase the overall readability and quality of the manuscript. At the moment apart from the occasional typos often the grammar is definitely suboptimal. Here are just some examples from the Introduction:

 “Keratoconus is a cone shaped which is bilateral corneal dystrophy which is progressive thinning corneal layer of stroma [1].”

“At the initial stage of the disease, there was no obvious signs appeared except mild loss in the visual acuity was observed in the patients. [4], severe signs like Fleischer’s ring, corneal protrusion, Vogt’s striae and Munson start to appear as disease progress from moderate to severe [4].” 

“Manifestation of the KC starts usually in 2nd decade of the life which will progress with its severity in third and fourth decades of life [10], while prevalence of this disease is more in males when compared to females in certain population [11, 12]. This disease prevalence pattern is more Sporadic oriented than familial cases while positive family history contribute to the onset of the disease in offspring. “

Many more examples are scattered among the text. The author will benefit from consultation with experienced and/or native English speaker in order to improve the grammar quality (and thus the overall clarity) of the text.

Author Response

Manuscript no: applsci-2292136

Manuscript title: Rare variants residing in novel cis-acting element in Visual system homeobox 1 (VSX1) and their contribution in the pathogenesis of Keratoconus.

Reviewer comment:

It will be good to compare (in terms of advantages and limitations) the cohort size and structure (number of patients and how they are distributed among the studied categories) in this study to the cohort size and structure in previously quoted studies on the topic (e.g. references 28-33). Such critical analysis may provide partial explanation of the discrepancies between the results reported by different teams.

Author’s response:

Authors appreciate the reviewer 1 for his kind words about the study and its outcome. We taken the idea of the reviewer 1 for better presentation of the manuscript by creating a Table no.5 in the manuscript and highlighting the comparison of the studies conducted on same pattern on different ethnicity, Study Type and cohort size, techniques, identified variants and number of patients. Based on the comparative analysis as suggested by the reviewer, the data inference is explained in detail in last paragraph of the discussion section of the revised manuscript.

Reviewer comment:

 The English spelling and grammar need thorough improvement which will increase the overall readability and quality of the manuscript. At the moment apart from the occasional typos often the grammar is definitely suboptimal. Here are just some examples from the Introduction:

“Keratoconus is a cone shaped which is bilateral corneal dystrophy which is progressive thinning corneal layer of stroma [1].”

“At the initial stage of the disease, there was no obvious signs appeared except mild loss in the visual acuity was observed in the patients. [4], severe signs like Fleischer’s ring, corneal protrusion, Vogt’s striae and Munson start to appear as disease progress from moderate to severe [4].” 

“Manifestation of the KC starts usually in 2nd decade of the life which will progress with its severity in third and fourth decades of life [10], while prevalence of this disease is more in males when compared to females in certain population [11, 12]. This disease prevalence pattern is more Sporadic oriented than familial cases while positive family history contribute to the onset of the disease in offspring. “

Many more examples are scattered among the text. The author will benefit from consultation with experienced and/or native English speaker in order to improve the grammar quality (and thus the overall clarity) of the text.

Author’s response:

Manuscript is rephrased and proof reading is done by the native English speaker who is also an eminent Emeritus Professor Edward J. Johns from University College Cork, Ireland. Manuscript with track changes highlight the number of changes done in new version of manuscript and a certificate is also attached about the manuscript proof reading.

Reviewer 2 Report

The paper investigates the role of VSX1 in keratoconus (KC) development in Pakistani families and sporadic cases. The study uses Sanger sequencing and in silico analysis to identify rare variants in VSX1 and their potential contribution to the development of KC. The results show a exon 3 synonymous polymorphism and a 3'UTR polymorphism in VSX1, which are not present in sporadic KC cases or control samples. In silico analysis reveals the potential role of these polymorphisms in KC development through the identification of novel cis-acting elements, including exonic splicing enhancer elements (ESE) and binding motifs for micro RNAs. The findings of the study suggest that rare genetic variations in VSX1 may contribute to the pathogenesis of KC through miRNA binding and regulation. The paper adds to the growing body of literature on the genetic basis of KC and provides a basis for future research into the role of VSX1 in KC development.

·      Although the method and results of the study presented in this paper are straightforward, the writing style is not easily readable. I suggest that the authors revise their writing to make it more accessible.

·      Furthermore, the authors suggest that the cis-acting element predicted in this study may be related to miRNA binding and regulation in KC disease. However, additional functional analysis is necessary to confirm this relationship.

·      Lastly, a few obvious typos were identified in the paper (as follows). Therefore, the authors should carefully proofread the paper to ensure that these errors are corrected.

1.     Line 67 double dots.

2.     Line 82-83 “and occupation (males=64, females=36) and occupation.”

3.     Line 94 “curvature” to “Curvature”.

4.     Line 108 “od” to “of”

5.     Line 115 format of -20 degree.

6.     Figure 1 legend numbers.

7.     Line 142 “Another program, another program”

8.     Line 156-158 redundant description to line 155-156.

9.     Line 208 SRSF5binding.

Author Response

Manuscript no: applsci-2292136

Manuscript title: Rare variants residing in novel cis-acting element in Visual system homeobox 1 (VSX1) and their contribution in the pathogenesis of Keratoconus.

Reviewer 2

Comments and Suggestions for Authors

The paper investigates the role of VSX1 in keratoconus (KC) development in Pakistani families and sporadic cases. The study uses Sanger sequencing and in silico analysis to identify rare variants in VSX1 and their potential contribution to the development of KC. The results show a exon 3 synonymous polymorphism and a 3'UTR polymorphism in VSX1, which are not present in sporadic KC cases or control samples. In silico analysis reveals the potential role of these polymorphisms in KC development through the identification of novel cis-acting elements, including exonic splicing enhancer elements (ESE) and binding motifs for micro RNAs. The findings of the study suggest that rare genetic variations in VSX1 may contribute to the pathogenesis of KC through miRNA binding and regulation. The paper adds to the growing body of literature on the genetic basis of KC and provides a basis for future research into the role of VSX1 in KC development.

 Reviewer comment:

Although the method and results of the study presented in this paper are straightforward, the writing style is not easily readable. I suggest that the authors revise their writing to make it more accessible.

Author’s response:

As per reviewer’s suggestions, the whole manuscript is read and corrected by a native English language speaker, Professor Emeritus Edward J. Johns, who is also an eminent Professor of Physiology in the University College Cork, to make it easily readable. His certificate of satisfactory English editing is attached.

Reviewer comment:

Furthermore, the authors suggest that the cis-acting element predicted in this study may be related to miRNA binding and regulation in KC disease. However, additional functional analysis is necessary to confirm this relationship.

Author’s response:

We appreciate the reviewer’s suggestion regarding the functional analysis of the interacting microRNA with novel cis-acting elements. Due to unavailability of the resources the functional analysis could not be performed, however we took another approach to highlight the probable functional role of microRNAs in manifestation of KC by incorporating the results of reported functional data of microRNAs by different groups. The available functional data has been incorporated in the revised manuscript and with citations listed below and explained in lines 357-354 in the last paragraph of the discussion section of the revised manuscript.

  1. Dong Q, Gu Y, Groome L, Wang Y. OS073. Over-expression of MIRNA-203 results in increased inflammatory response in endothelial cells: a mechanism of increased endothelial inflammatory response in preeclampsia. Pregnancy Hypertens. 2012 Jul;2(3):217..
  2. Reyhan AH, Karadağ AS, Çınar ŞŞ. Assessing the role of systemic inflammation in the etiopathogenesis of advanced stage keratoconus. Indian J Ophthalmol. 2021 Oct;69(10):2658-2662.
  3. Nichani PAH, Solomon B, Trinh T, Mimouni M, Rootman D, Singal N, Chan CC. Investigating the role of inflammation in keratoconus: A retrospective analysis of 551 eyes. Eur J Ophthalmol. 2023 Jan;33(1):35-43.

Reviewer comment:

Lastly, a few obvious typos were identified in the paper (as follows). Therefore, the authors should carefully proofread the paper to ensure that these errors are corrected.

  1.      Line 67 double dots.
  2. Line 82-83 “and occupation (males=64, females=36) and occupation.”
  3. Line 94 “curvature” to “Curvature”.
  4. Line 108 “od” to “of”
  5. Line 115 format of -20 degree.
  6. Figure 1 legend numbers.
  7. Line 142 “Another program, another program”
  8. Line 156-158 redundant description to line 155-156.
  9. Line 208 SRSF5binding.

Author’s response:

In addition to above mentioned points, manuscript was corrected at many places which can be seen in track changes version of the manuscript. Whole manuscript is read and corrected by a native English language speaker, Professor Emeritus Edward J. Johns, who is also an eminent Professor of Physiology in the University College Cork, to make it easily readable. His certificate of satisfactory English editing is attached.

Round 2

Reviewer 2 Report

The paper investigates the role of VSX1 in keratoconus (KC) development in Pakistani families and sporadic cases. The study uses Sanger sequencing and in silico analysis to identify rare variants in VSX1 and their potential contribution to the development of KC. The results show a exon 3 synonymous polymorphism and a 3'UTR polymorphism in VSX1, which are not present in sporadic KC cases or control samples. In silico analysis reveals the potential role of these polymorphisms in KC development through the identification of novel cis-acting elements, including exonic splicing enhancer elements (ESE) and binding motifs for micro RNAs. The findings of the study suggest that rare genetic variations in VSX1 may contribute to the pathogenesis of KC through miRNA binding and regulation. The paper adds to the growing body of literature on the genetic basis of KC and provides a basis for future research into the role of VSX1 in KC development.

Please pay attention to entire paper grammar/typo like line 109 two “from”. 

Author Response

Revision 2

Manuscript no: applsci-2292136

Manuscript title: Rare variants residing in novel cis-acting element in Visual system homeobox 1 (VSX1) and their contribution in the pathogenesis of Keratoconus.

Reviewer 2

Comments and Suggestions for Authors

The paper investigates the role of VSX1 in keratoconus (KC) development in Pakistani families and sporadic cases. The study uses Sanger sequencing and in silico analysis to identify rare variants in VSX1 and their potential contribution to the development of KC. The results show a exon 3 synonymous polymorphism and a 3'UTR polymorphism in VSX1, which are not present in sporadic KC cases or control samples. In silico analysis reveals the potential role of these polymorphisms in KC development through the identification of novel cis-acting elements, including exonic splicing enhancer elements (ESE) and binding motifs for micro RNAs. The findings of the study suggest that rare genetic variations in VSX1 may contribute to the pathogenesis of KC through miRNA binding and regulation. The paper adds to the growing body of literature on the genetic basis of KC and provides a basis for future research into the role of VSX1 in KC development.

Please pay attention to entire paper grammar/typo like line 109 two “from”. 

Author’s response:

We appreciate the comments of reviewer 2 to give us a chance to further refine the manuscript by removing typo, grammatical and spelling mistakes. According to reviewer’s suggestions, the whole manuscript is read and corrected for typo, grammatical and spelling mistakes. Revised version of manuscript is corrected at approximately 38-40 places which can be seen in pdf version having track changes. We expect to meet the need of applied Science journal and satisfaction of the reviewer and editor about the publication of this manuscript.
